# INAVA-ARNO complexes bridge mucosal barrier function with inflammatory signaling

Phi Luong[1,2], Matija Hedl[3], Jie Yan[3], Tao Zuo[1,2], Tian-Min Fu[4,5], Xiaomo Jiang[6], Jay R Thiagarajah[1,2,7], Steen H Hansen[1,2,7], Cammie F Lesser[8,9], Hao Wu[4,5], Clara Abraham[3]*, Wayne I Lencer[1,2,7]*

[1]Division of Gastroenterology, Nutrition and Hepatology, Boston Children's Hospital, Boston, United States; [2]Department of Pediatrics, Harvard Medical School, Boston, United States; [3]Department of Medicine, Yale University, New Haven, United States; [4]Department of Biological Chemistry and Molecular Pharmacology, Harvard Medical School, Boston, United States; [5]Program in Cellular and Molecular Medicine, Boston Children's Hospital, Boston, United States; [6]Novartis Institutes for Biomedical Research, Cambridge, United States; [7]Harvard Digestive Disease Center, Harvard Medical School, Boston, United States; [8]Department of Medicine, Division of Infectious Diseases, Massachusetts General Hospital, Cambridge, United States; [9]Department of Microbiology and Immunobiology, Harvard Medical School, Boston, United States

**\*For correspondence:**
clara.abraham@yale.edu (CA);
Wayne.Lencer@childrens.harvard.edu (WIL)

**Abstract** Homeostasis at mucosal surfaces requires cross-talk between the environment and barrier epithelial cells. Disruption of barrier function typifies mucosal disease. Here we elucidate a bifunctional role in coordinating this cross-talk for the inflammatory bowel disease risk-gene *INAVA*. Both activities require INAVA's DUF3338 domain (renamed CUPID). CUPID stably binds the cytohesin ARF-GEF ARNO to effect lateral membrane F-actin assembly underlying cell-cell junctions and barrier function. Unexpectedly, when bound to CUPID, ARNO affects F-actin dynamics in the absence of its canonical activity as a guanine nucleotide-exchange factor. Upon exposure to IL-1β, INAVA relocates to form cytosolic puncta, where CUPID amplifies TRAF6-dependent polyubiquitination and inflammatory signaling. In this case, ARNO binding to CUPID negatively-regulates polyubiquitination and the inflammatory response. INAVA and ARNO act similarly in primary human macrophages responding to IL-1β and to NOD2 agonists. Thus, INAVA-CUPID exhibits dual functions, coordinated directly by ARNO, that bridge epithelial barrier function with extracellular signals and inflammation.
DOI: https://doi.org/10.7554/eLife.38539.001

## Introduction

C1ORF106, recently named INAVA (Innate Immune Activator), was identified as a risk factor for the chronic inflammatory bowel diseases (IBD) by genome-wide association studies and targeted exome sequencing (*Rivas et al., 2011*). Mice lacking the protein altogether show defects in intestinal barrier integrity at steady state and greater susceptibility to mucosal infection (*Mohanan et al., 2018*). Human macrophages carrying the IBD rs7554511 risk allele have decreased INAVA expression and show multiple defects in myeloid function, including in innate immune NOD2 signaling and cytokine secretion, and in microbial clearance in association with reduced autophagy and ROS production (*Yan et al., 2017*). Each process is well known to affect gut function in health and disease, but the

molecular mechanisms for how they are regulated or interconnected by INAVA are not fully understood.

We previously determined that INAVA is strongly enriched in simple epithelial cells (*Nelms et al., 2016*) - the cell type that forms mucosal barriers. By domain analysis, the molecule has a single distinguishing feature, the Domain of Unknown Function DUF3338 (which we rename CUPID for Cytohesin Ubiquitin Protein Inducing Domain). Three other human proteins contain CUPID: FRMD4a, FRMD4B, and CCDC120, and two are implicated in human disease (*Cappola et al., 2010*; *Fine et al., 2015*; *Garner et al., 2014*; *Goldie et al., 2012*; *Lambert et al., 2013*; *Velcheti et al., 2017*; *Yoon et al., 2012*). All appear to bind the ARF-GEF (guanine nucleotide-exchange factors) cytohesin family members (*Huttlin et al., 2017*; *Ikenouchi and Umeda, 2010*; *Klarlund et al., 2001*; *Mohanan et al., 2018*; *Torii et al., 2014*).

The cytohesins are guanine nucleotide-exchange factors for the ARF-family of proteins (ARF 1–4), which regulate cell membrane and F-actin dynamics (*Donaldson and Jackson, 2011*; *Stalder and Antonny, 2013*). All cytohesins contain a N-terminal coiled-coil (CC) protein-protein interaction region, an enzymatic SEC7 guanine nucleotide-exchange factor (GEF) domain, and a C-terminal PIP-binding PH domain. In their inactive conformation, the cytohesins localize to the cytosol. Full-blown GEF activation, typified by cytohesin 2 (also known as ARNO), requires membrane recruitment via ARNO binding to PIP2 (phosphatidylinositol 4, 5-bisphosphate), and then (activated) ARF-GTP, a product of the ARNO-GEF reaction (*Chardin et al., 1996*; *Cohen et al., 2007*; *Malaby et al., 2013*). This enables an enzymatically-driven positive feedback-loop for rapidly amplifying a localized pool of activated cytohesins and ARF-GTP needed to drive the massive ARF-dependent changes in actin and membrane dynamics that underlie cell spreading and epithelial breakdown (*Santy and Casanova, 2001*; *Stalder et al., 2011*).

In this study, we address the mechanism of INAVA action in polarized intestinal epithelial cells and primary human macrophages. We discover dual and mutually-exclusive functions for INAVA and the physical and functional interaction of the INAVA CUPID domain (INAVA-CUPID) with cytohesin 2 ARNO. In epithelial cells, INAVA-CUPID recruits ARNO to lateral membranes where the complex promotes actin assembly that underlies barrier function. This occurs via a novel GEF activity-independent mechanism. In response to the inflammatory cytokine IL-1β, INAVA relocates to cytosolic puncta that function as signalosomes. Here, CUPID acts with the E3-ubiquitin-ligase TRAF6 to enhance inflammatory signaling, and in this case, ARNO binding inhibits CUPID activity. In human macrophages containing the INAVA rs7554511 IBD-risk allele (low-INAVA expressing carriers), wild type INAVA expression enhances, and ARNO expression suppresses IL-1β and NOD2 signaling. Reconstitution with purified proteins in vitro shows biochemically that INAVA-CUPID functions as an enhancer of TRAF6 dependent polyubiquitination, and that this is blocked by ARNO. These results provide a direct mechanistic link between mucosal barrier function and inflammation implicated in human disease.

## Results

### INAVA affects the epithelial barrier

To investigate the function of INAVA, we first generated INAVA shRNA knockdown and CRISPR knockout Caco2BBe human intestinal cells (*Figure 1—figure supplement 1A,B*). Caco2BBe cells lacking INAVA show enhanced cell spreading, while cells stably expressing INAVA-GFP are similar to wild type (*Figure 1A*; *Figure 1—figure supplement 1A,C*). When grown on permeable supports as polarized monolayers, INAVA-deficient cells display increased permeability to 4 kDa FITC dextran and reduced transepithelial electrical resistance (TEER) (*Figure 1—figure supplement 1D–F*), implicating defects in paracellular permeability and intercellular junctions. Conversely, Caco2BBe monolayers that overexpress INAVA exhibit evidence of increased integrity of intercellular junctions as measured by higher TEER and greater tolerance to depletion of intercellular adherens junctions caused by exposure to low $Ca^{2+}$ (*Figure 1—figure supplement 1G*). These phenotypes are consistent with a recent study using INAVA knock-out mice that supports a role for INAVA in maintenance of intercellular epithelial junctions responsible for mucosal barrier function (*Mohanan et al., 2018*).

To understand how INAVA acts on epithelial junctions, we first examined the localization of INAVA constructs fused to GFP that were stably expressed in Caco2BBe cells. Humans express two

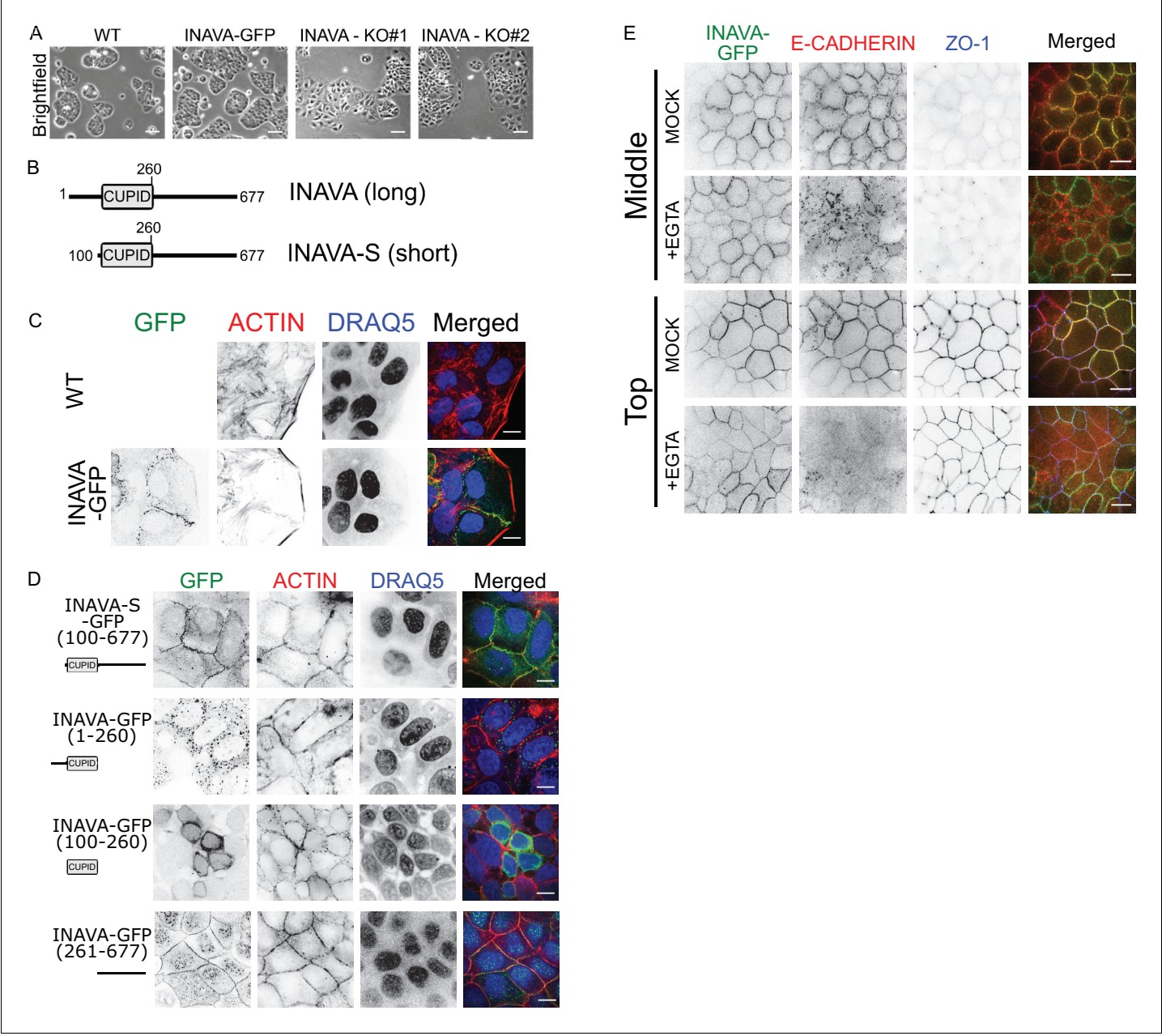

**Figure 1.** INAVA localizes to cell-cell junctions. (**A**) Domain architecture of the long and short INAVA isoforms.(**B**) Brightfield images of Caco2BBe WT, INAVA-GFP over-expressing, and INAVA CRISPR knockout cells grown 2 days on coverslips. Scalebar = 200 μm. (**C**) Localization of stably expressed long isoform INAVA-GFP in Caco2BBe grown on coverslips. Cells stained with F-actin (TRITC-phalloidin) and nuclei (DRAQ5). (**D**) Localization of stably expressed GFP tagged INAVA constructs in Caco2BBe cells.(**E**) Localization of Caco2BBe polarized epithelial monolayers stably expressing INAVA-GFP on 2-D transwells. Cells were treated for 5 min with 6 mM EGTA and then immediately fixed with 4% paraformaldehyde and stained for E-cadherin (adherens junction, red) and ZO-1 (tight junction, blue). Scalebar 10 μm (see also *Figure 1—figure supplement 1I,J*).

DOI: https://doi.org/10.7554/eLife.38539.002

The following figure supplement is available for figure 1:

**Figure supplement 1.** INAVA is important for barrier function in epithelial Caco2BBe cells.

DOI: https://doi.org/10.7554/eLife.38539.003

isoforms of INAVA. The short-isoform (INAVA-S) lacks a 99-amino-acid segment N-terminal to CUPID (*Figure 1B*). Both long INAVA-GFP and short INAVA-S-GFP isoforms localized to cell-cell contacts (*Figure 1C,D*; *Figure 1—figure supplement 1H*). Notably, leading edges of migratory cells were uniformly devoid of INAVA. In well-differentiated polarized Caco2BBe monolayers, INAVA localized to tight and adherens junctions, as evidenced by co-localization with E-cadherin and ZO-1 (*Figure 1—figure supplement 1I,J*). When expressed on its own, the C-terminal region flanking the INAVA-CUPID domain (aa 261–677) also localized to lateral membranes (*Figure 1D*, bottom panels). The CUPID domain on its own localized to the cytosol, and the CUPID domain together with the N-terminal flanking region (aa 1–260) localized to cytosolic puncta (*Figure 1D*). Thus, the C-terminal region of INAVA is necessary and sufficient for lateral membrane targeting. Lateral membrane localization did not depend upon association with the E-cadherin complex. This was evidenced in cells treated with 6 mM EGTA to disassemble adherens junctions and induce E-cadherin endocytosis. While E-cadherin localized to endocytic vesicles upon EGTA treatment, INAVA remained at lateral membranes (*Figure 1E*). Staining for the tight junction protein ZO-1 as a control also remained unchanged. Thus, INAVA localizes to lateral membranes independently of E-cadherin. Overall, these results support a critical role for INAVA in maintaining or regulating epithelial barrier integrity.

## CUPID-domain directly interacts with cytohesin family members

To understand the mechanism of action underlying this phenotype, we focused on the single signature feature of INAVA, the CUPID domain. As all four CUPID-containing human proteins are implicated in binding the cytohesins (*Ikenouchi and Umeda, 2010*; *Klarlund et al., 2001*; *Mohanan et al., 2018*; *Torii et al., 2014*), we tested if cytohesin-binding is involved in INAVA function and explained by CUPID. We first screened the cytohesins 1–3 for binding to INAVA-CUPID (aa 100–261) using a yeast-based protein-protein interaction platform (*Schmitz et al., 2009*). In all cases, as evidenced by intracellular puncta formation, CUPID assembled with each of the cytohesins: cytohesin 1, cytohesin 2 (ARNO), and cytohesin 3 (GRP1), (*Figure 2—figure supplement 1A,B*). Coiled-coil domains of the cytohesins were sufficient for direct interaction with CUPID. In a purified system, recombinant INAVA-CUPID was sufficient to pull down the coiled-coil domains of Cytohesin-1, ARNO, and GRP1 (*Figure 2A*; *Figure 2—figure supplement 1C*). Thus, cytohesin binding by INAVA is direct and dependent upon CUPID.

CUPID binding to the coiled-coil regions of the cytohesins, however, did not affect their GEF activity, as assessed biochemically using recombinant proteins in vitro (*Figure 2—figure supplement 1D*). To test how INAVA may affect cytohesin function, we generated Caco2BBe intestinal cells stably expressing myc-tagged ARNO or GRP1. These cells, however, rapidly suppressed ARNO and GRP1 expression, as evidenced by the lack of protein detected upon transduction. PMA (phorbol-12-myristate-13-acetate), which strongly activates the CMV promotors used in these transductions, rescued ARNO and GRP1 protein expression, presumably by overcoming the endogenous mechanism(s) of epigenetic gene silencing operating in these cell lines (*Figure 2B*, compare lanes 1 and 3; and lanes 5 and 7) (*Löser et al., 1998*). Strikingly, when co-expressed in the same cells, INAVA rescued ARNO expression in the absence of PMA (*Figure 2B*, compare lanes 1 and 2), implicating a stabilizing interaction between the two proteins. The effect was specific to ARNO as INAVA did not rescue the cytohesin GRP1 against gene-suppression (compare lanes 5 and 6). Over-expression of INAVA-GFP increased endogenous ARNO protein levels with no apparent changes in TEER or paracellular flux when compared to wildtype cells (*Figure 1—figure supplement 1E,F*; *Figure 2—figure supplement 1E*). Two independently created INAVA KO Caco2BBe cell lines showed no evidence for increased ARNO levels (*Figure 2—figure supplement 1F*), unlike the results recently reported for INAVA's effect on cytohesin 1, a closely related ARF-GEF member of the cytohesin family. In accord with these results, we found durable expression of ARNO in primary human macrophages over-expressing INAVA (*Figure 2—figure supplement 1G*). Thus, INAVA interacts with and appears to stabilize ARNO expression in intact cells, consistent with our biochemical data, but in contrast to a recent report that INAVA induces the ubiquitin-dependent degradation of the cytohesins (*Mohanan et al., 2018*).

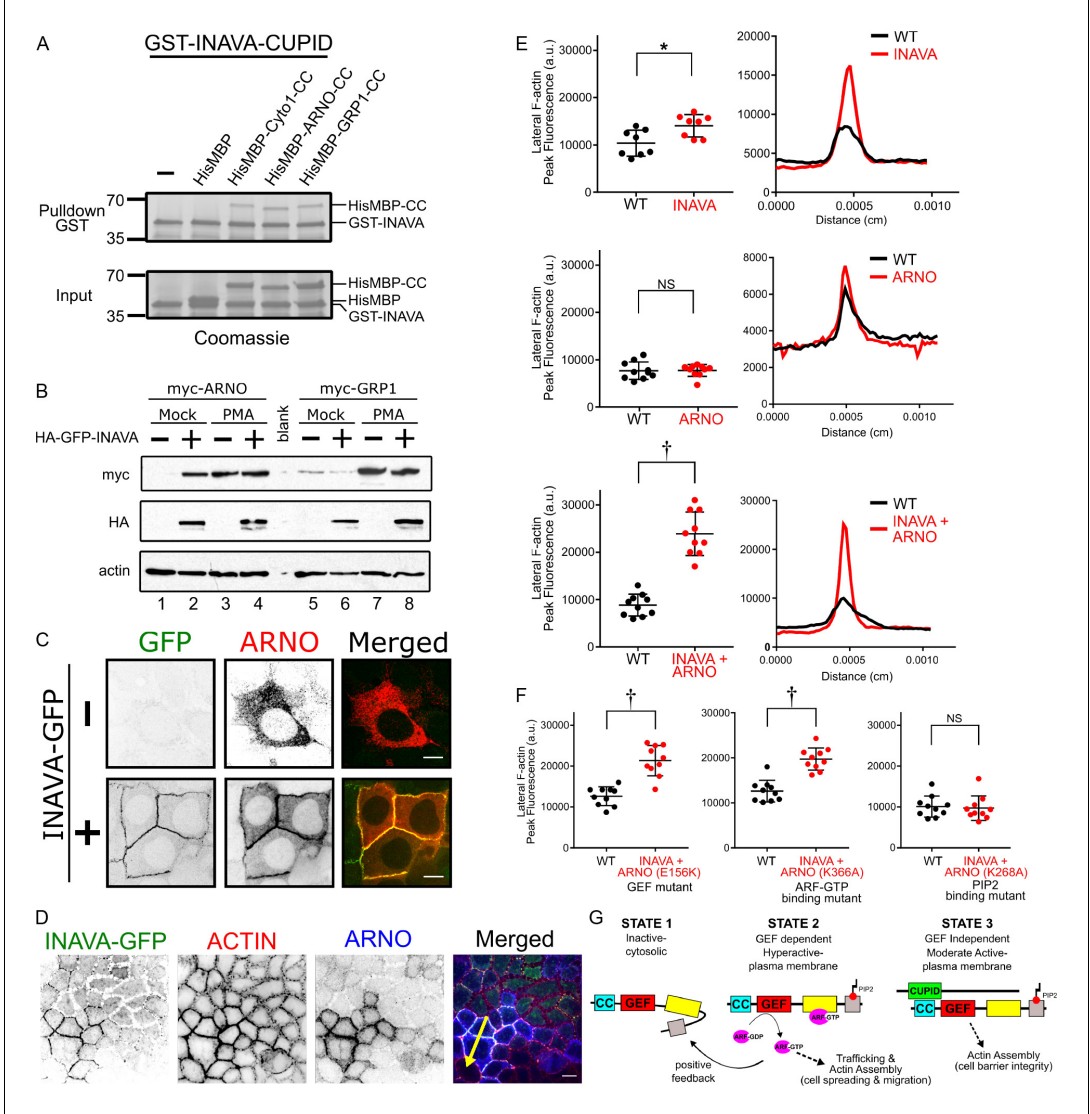

**Figure 2.** INAVA recruits ARNO to cell-cell junctions to promote GEF-independent actin assembly. (**A**) GST pulldowns with recombinant GST-INAVA-CUPID domain and His-MBP-CC (coiled-coil) of cytohesin members (see also *Figure 2—figure supplement 1C*). (**B**) Immunoblot of Caco2BBe stably transduced with lentiviral vector driving myc-ARNO and myc-GRP1 expression under constitutive CMV promoter. Cells were additionally transduced with lentivirus expressing HA-GFP-INAVA or treated with phorbol-12-myristate-13-acetate (PMA). (**C**) Confocal microscopy images of Caco2BBe stably expressing doxycycline inducible myc-ARNO alone or with constitutively expressed INAVA-GFP. Scalebar = 10 μm. (**D**) Confocal images of coculture of Caco2BBe WT and stably expressing INAVA-GFP with myc-ARNO. Cells were stained with (**E**) Quantification of peak fluorescence intensity of cell-cell junction F-actin line scans as in (**F**), from Caco2BBe WT or Caco2BBe stably expressing INAVA-GFP (top), myc-ARNO (middle), or INAVA-GFP coexpressed with myc-ARNO (bottom). Right panel shows representative profiles of F-actin line scans. (**F**) F-actin line scans with Caco2BBe cocultured with Caco2BBe co-expressing INAVA-GFP and mutant ARNO constructs including GEF mutant (E156K), ARF6-GTP mutant (K366A), PIP2 mutant (K268A). Error bars indicate ± SEM. *p<0.05, †p<0.0001, NS not significant (two-tailed Student's t-test), n = 8–10. (**G**) Cartoon display domain architecture of INAVA and ARNO and three ARNO activity states. Inactive ARNO is autoinhibited and cytosolically localized. The GEF dependent positive feedback mechanism of ARNO promotes robust levels of trafficking and actin assembly at the plasma membrane. In the context of a polarized epithelial cells this positive feedback mechanism disrupts barrier integrity that leads to epithelial breakdown and cell spreading. In contrast, CUPID domain mediated INAVA-ARNO complex permits non-canonical GEF independent function that allow fine control of lateral actin to maintain barrier homeostasis.

DOI: https://doi.org/10.7554/eLife.38539.004

The following figure supplement is available for figure 2:

**Figure supplement 1.** Cytohesins interact with INAVA to Promote GEF Independent Actin Assembly.

DOI: https://doi.org/10.7554/eLife.38539.005

## INAVA-ARNO promotes non-canonical GEF-independent lateral membrane actin assembly

To circumvent the problem of gene silencing in cells stably expressing ARNO and enable further mechanistic studies on INAVA-ARNO interactions, we generated stable doxycycline (dox) inducible myc-tagged ARNO expressing Caco2BBe cells. Upon induction with dox, ARNO localized to the cytosol, consistent with the molecule folding in an auto-inhibited inactive state (*Figure 2C* top panels) (*Cohen et al., 2007*). Co-expression of INAVA-GFP recruited ARNO to lateral membranes, consistent with our findings that ARNO and INAVA form stable protein complexes (*Figure 2C* bottom panels). Unexpectedly, given ARNO's canonical actions in regulating F-actin assembly and membrane dynamics (*Turner and Brown, 2001*), the INAVA-ARNO complex did not cause epithelial cell spreading or disassembly of intercellular contacts. Intercellular junctions remained fully intact as assessed by TEER (*Figure 1—figure supplement 1F*). We did, however, find enhancement of lateral membrane F-actin assembly (*Figure 2D,E*), consistent with the anticipated effects of ARF activation on cortical F-actin (*Myers and Casanova, 2008*). In these studies, to properly compare the densities of cortical F-actin, we co-cultured cells stably expressing INAVA, or ARNO, or both proteins, alongside non-transduced wild type (WT) Caco2BBe cells (*Figure 2D*). Relative densities of lateral membrane F-actin stained with TRITC-phalloidin was measured by imaging both cell types on the same cover slip under the same conditions (*Figure 2E* right panels, peak intensities plotted in left panels). Cells expressing INAVA alone showed modest amplified F-actin density at lateral membranes. Expression of ARNO alone had no detectable effect. In contrast, expression of both INAVA and ARNO together strongly amplified lateral membrane cortical F-actin (>2 fold). Remarkably, the effects of INAVA-ARNO on F-actin assembly did not require ARF-GTP binding to ARNO or ARNO's enzymatic GEF activity (*Figure 2F*), as we observed the same enhanced densities of lateral membrane F-actin in Caco2BBe cells expressing the ARNO GEF mutant (E156K) or the ARF-GTP binding mutant (K366A). In contrast, ARNO mutants lacking the ability to bind PIP$_2$ (K268A) failed to enhance F-actin (*Figure 2F*, right). Thus, in intact cells, when complexed with INAVA, only PIP binding by ARNO is required for effecting F-actin assembly.

To identify the INAVA region required for ARNO-mediated F-actin effects, we examined cortical F-actin assemblies in Caco2BBe cells expressing ARNO together with just the INAVA-CUPID domain. In this case, because CUPID alone cannot target ARNO to lateral membranes (*Figure 1D*), we found evidence for increased F-actin on intracellular vesicles, where both CUPID and ARNO co-localized (*Figure 2—figure supplement 1H*). This phenotype was amplified and more easily observed in human mammary epithelial MCF7 cells. In MCF7 cells, both INAVA and ARNO colocalized on intracellular vesicles, and both proteins were required for the induction of F-actin rich vesicles. Expression of each protein alone did not induce the vesicle phenotype (*Figure 2—figure supplement 1I,J*). The ARNO-GEF (E156K) and ARF-GTP binding (K336A) mutants also promoted formation of F-actin coated vesicles, consistent with our earlier results. In contrast, as in the Caco2BBe model, INAVA and the ARNO PIP$_2$-binding mutant did not promote actin assembly (*Figure 2—figure supplement 1K*). Thus, the GEF and ARF-binding activities of ARNO are dispensable (*Figure 2F*; *Figure 2—figure supplement 1K*), again implicating a unique and non-canonical mechanism of action for the cytohesins in regulating F-actin dynamics when bound to INAVA. In other words, instead of the substantial ARNO GEF-dependent hyperactivation of ARFs leading to massive F-actin rearrangement, cell spreading, and epithelial breakdown (*Stalder and Antonny, 2013*; *White et al., 2010*), CUPID binding to ARNO displaces the positive-feedback hyperactivation of ARFs to enable instead a smaller (modulated) effect on cortical F-actin assembly, which we propose is required for barrier integrity (*Figure 2G*). Canonically, activated ARNO localizes to the leading edge to drive cell motility, but here we show INAVA localizes ARNO to lateral membranes, where regulation of cell-cell contacts and mucosal barrier function occurs.

## INAVA-ARNO regulates IL-1β signaling

ARNO has been implicated in IL-1β signaling (*Zhu et al., 2012*), and given the association of INAVA with inflammatory bowel disease, we examined the effect of IL-1β on INAVA-ARNO function. We found that IL-1β rapidly within 30 min induced the relocalization of INAVA-GFP from lateral membranes to cytosolic puncta in Caco2BBe cells (*Figure 3A*). IL-1β-induced cytosolic puncta were also found in human colon HCT8 cells where we found a stronger phenotype (*Figure 3B*). In this case,

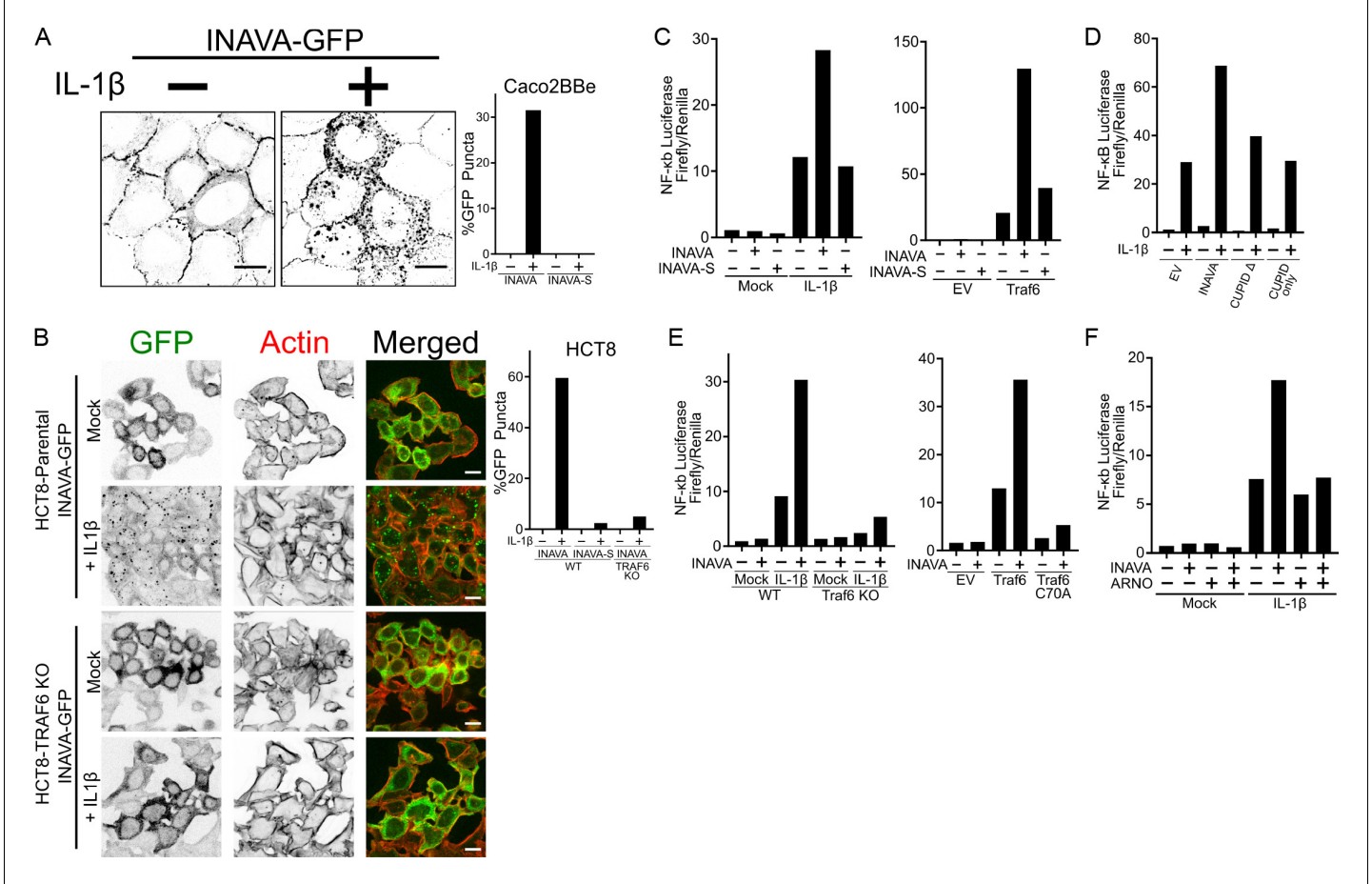

**Figure 3.** INAVA is an activator of IL-1β signaling and is inhibited by ARNO. (**A and B**) Confocal images of (**A**) Caco2BBe and (**B**) HCT8 or HCT8 TRAF6 KO cells stably expressing INAVA-GFP long isoform treated with or without 10 ng/ml IL-1β for 30 min. Graph display quantification of %GFP positive puncta of long and short INAVA, n = 73–125 cells. Scalebar = 10 μm (see also *Figure 3—figure supplement 1A,B*). (**C**) NF-κB reporter assays in HEK293T transfected with INAVA long or short INAVA-S and stimulated with 10 ng/ml IL-1β for 4 hr, left, or co-transfected with TRAF6, right graph. Reporter data is average of duplicates and representative of three independent experiments. (**D**) As in (**C**) but with long INAVA isoform, INAVA without CUPID domain and CUPID only. (**E**) As in (**C**) but with TRAF6 knockout, left graph and TRAF6 ligase mutant C70A, right graph (see also *Figure 3—figure supplement 1C*). (**F**) As in (**C**) but with stably expressed ARNO.

DOI: https://doi.org/10.7554/eLife.38539.006

The following source data and figure supplement are available for figure 3:

**Source data 1.** Source Data for Figure 3C.
DOI: https://doi.org/10.7554/eLife.38539.007
**Source data 2.** Source Data for Figure 3D.
DOI: https://doi.org/10.7554/eLife.38539.008
**Source data 3.** Source Data for Figure 3E.
DOI: https://doi.org/10.7554/eLife.38539.009
**Source data 4.** Source Data for Figure 3F.
DOI: https://doi.org/10.7554/eLife.38539.010
**Figure supplement 1.** INAVA-GFP puncta formation is relatively less in HCT8 cells stimulated with TNFα.
DOI: https://doi.org/10.7554/eLife.38539.011

puncta were assembled from cytosolic pools of INAVA-GFP as HCT8 cells do not form intercellular junctions (*Vermeulen et al., 1997*) where INAVA-GFP normally localizes. INAVA puncta appeared to be associated with IL-1β signaling as IL-1β-induced INAVA-puncta were not efficiently formed in HCT8 cells lacking TRAF6 (CRISPR-Cas9 KO), the ubiquitin E3 ligase required for IL-1β signal transduction (*Figure 3B*; *Figure 3—figure supplement 1A*) (*Cao et al., 1996*; *Lomaga et al., 1999*).

TNFα, another inflammatory cytokine, induced INAVA puncta to a much lower extent (*Figure 3—figure supplement 1B*), suggesting selectivity for INAVA action in different inflammatory cascades. Notably, the short INAVA-S isoform rarely induced puncta formation in Caco2BBe or HCT8 intestinal cells (*Figure 3A,B*). We conclude the N-terminal region of INAVA is required for puncta formation.

To test if INAVA affects IL-1β signaling, we used HEK293T cells expressing INAVA together with a NF-κB-dependent luciferase reporter. The cells were treated with IL-1β or co-transfected with TRAF6, which also induces an inflammatory response. Cells expressing INAVA responded to IL-1β or TRAF6 overexpression by enhanced NF-κB activation, as assessed using luciferase expression (*Figure 3*; numerical values *Figure 3—source data 1*). Cells expressing the short INAVA-S isoform, however, did not. This result aligns with our finding that the short isoform cannot assemble into cytosolic puncta, further implicating puncta formation in the signaling cascade. We also found that cells expressing INAVA lacking the CUPID domain (CUPID Δ) or cells expressing only the CUPID domain failed to show amplified IL-1β signal transduction (*Figure 3D*; numerical values *Figure 3—source data 2*). Thus, the CUPID domain is required, but not sufficient.

Based on these studies, we conclude INAVA-puncta mark the formation of multiprotein signalosomes required for affecting inflammatory responses. Though INAVA is not essential for IL-1β induced signaling, as further evidenced in Caco2BBe INAVA KO cells lacking INAVA entirely (*Figure 3—figure supplement 1C*), INAVA is required for amplifying the inflammatory response. The signalosomes mediating this activity depend on the INAVA-N-terminal and CUPID domains (*Figure 3D*; numerical values *Figure 3—source data 2*), and on the enzymatic activity of TRAF6. This is best evidenced in HEK293T cells lacking TRAF6, or in cells over-expressing its enzymatic mutant (C70A). Both cell lines failed to mediate the enhancement of IL-1β signal transduction (*Figure 3E*; *Figure 3—figure supplement 1D*; numerical values *Figure 3—source data 3*). Remarkably, we also found that ARNO over-expression blocked INAVA's activity in IL-1β signal transduction (*Figure 3F*; numerical values *Figure 3—source data 4*), presumably by tightly binding the INAVA CUPID domain and interfering with signalosome function. This result uncovers a second previously unappreciated function for ARNO as a negative regulator of INAVA-dependent inflammatory cascades.

## INAVA-ARNO amplifies inflammatory signaling in human macrophages

To further address the relevance of these findings, we prepared primary monocyte-derived human macrophages (MDMs) from individuals carrying the INAVA-IBD rs7554511 CC risk allele. These cells have reduced INAVA expression and impaired signaling and cytokine secretion in response to PRR stimulation (*Yan et al., 2017*). We asked if similar genotype-dependent regulation was observed in response to IL-1β treatment. As predicted from our previous results (*Yan et al., 2017*), IL-1β-treated MDMs from rs7554511 CC IBD risk carriers showed reduced INAVA protein expression compared to MDMs derived from AA carriers (*Figure 4A*). Also consistent with our in vitro results, both IL-1β-induced NF-κB and MAPK signaling (*Figure 4B*), and cytokine secretion (*Figure 4C*), were reduced in MDMs from rs7554511 CC IBD risk carriers. To then define the role of the CUPID domain of INAVA in human MDMs, we transfected MDMs from rs7554511 CC IBD risk carriers (low INAVA-expressing carriers) with the various INAVA constructs (*Figure 4—figure supplement 1A–B*); cell survival was unchanged under these transfected conditions (*Figure 4—figure supplement 1C*). Compared to MDMs transfected with empty vector (EV), MDMs expressing wild type INAVA responded to IL-1β or MDP treatment by robust activation (phosphorylation) of NF-κB and the MAP-kinases ERK, p38, and JNK (*Figure 4D,E*, black bars; *Figure 4—figure supplement 1D*). The same results were obtained when measuring IL-1β- and MDP-induced cytokine release (*Figure 4F, G*). Cells expressing INAVA lacking the CUPID domain (CUPID Δ), or MDMs expressing CUPID only, failed to respond, as compared to empty vector controls (*Figure 4D,E*, grey bars; *Figure 4F,G*). We also found that in all cases the expression of ARNO (*Figure 4—figure supplement 1E–G*) blocked the IL-1β- and MDP-induced, INAVA-dependent NF-κB and MAPK activation and TNF secretion in a dose-dependent manner (*Figure 4H*, black bars; *Figure 4I*; *Figure 4—figure supplement 1H,I*). Thus, primary human macrophages also exhibit INAVA-ARNO mediated regulation of IL-1β (and MDP) signaling. Moreover, as in unstimulated MDMs, INAVA did not reduce ARNO expression under inflamed conditions (*Figure 4—figure supplement 1F*), consistent with our findings that INAVA stabilizes rather than degrades ARNO.

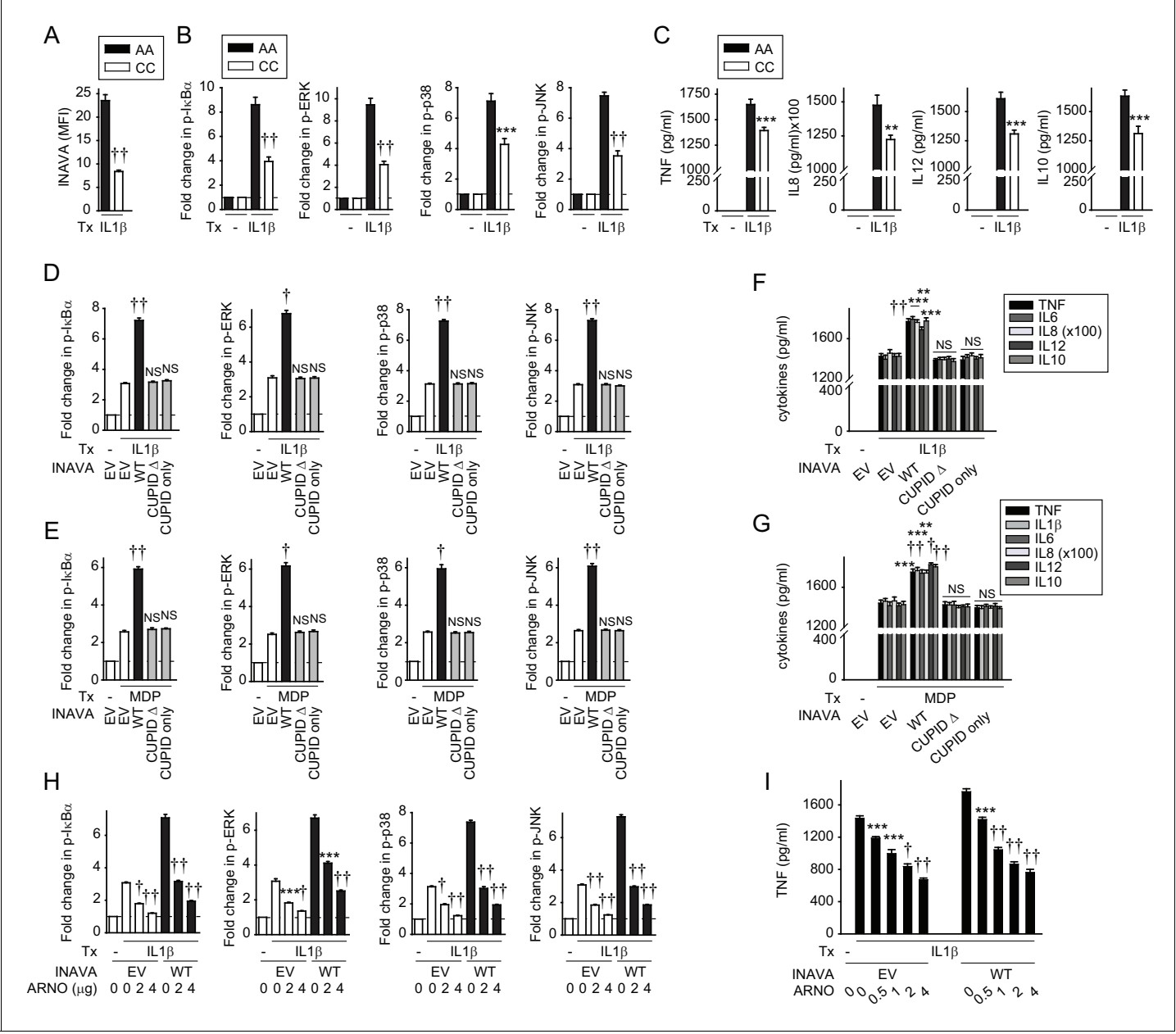

**Figure 4.** IL-1β-initiated NFκB and MAPK signaling and cytokine secretion is decreased in MDMs from INAVA rs7554511 CC risk carriers. (**A–C**) MDMs from rs7554511 AA (INAVA high-expressing) and CC (INAVA low-expressing) (n = 10/genotype) carriers were treated with 10 ng/ml IL-1β. (**A**) Summary graph of INAVA expression as assessed by flow cytometry. Mean fluorescent intensity (MFI) + SEM. (**B**) Summary graphs of fold MFI of the indicated phospho-kinases + SEM. (**C**) Mean cytokine secretion at 24h + SEM. Significance is compared to IL-1β-treated, rs7554511 AA carrier MDMs. (**D–G**) MDMs from rs7554511 CC homozygote (low-expressing) INAVA carriers were transfected with empty vector (EV) or the indicated HA-INAVA constructs. Cells were treated with (**D and F**) 10 ng/ml IL-1β or (**E and G**) 100 μg/ml MDP. (**D and E**) Summary graphs of MFI fold change of the indicated phospho-proteins at 15min + SEM (n = 6, similar results seen in an additional n = 10 for (**E**)). (**F and G**) Mean cytokine secretion at 24 hr + SEM (n = 6, similar results seen in an additional n = 10 for (**G**)) (see also *Figure 4—figure supplement 1A-D*). (**H and I**) As in (**D and F**) but with Myc-ARNO vector at the indicated concentrations. (**H**) Summary graphs with MFI fold change of indicated phospho-proteins at 15 min +SEM. (**I**) Mean TNF secretion at 24 hr + SEM. Significance is compared to: treated, EV-transfected cells for (**D–G**), and treated cells without ARNO transfection (EV) for each respective condition for (**H and I**). Tx, treatment; EV, empty vector, NS, not significant; **, p<0.01; ***, p<0.001; †, p<1×10$^{-4}$; ††, p<1×10$^{-5}$.

DOI: https://doi.org/10.7554/eLife.38539.012

The following figure supplement is available for figure 4:

**Figure supplement 1.** ARNO inhibits INAVA-dependent NOD2-induced signaling and cytokine secretion.

DOI: https://doi.org/10.7554/eLife.38539.013

## INAVA enhances protein ubiquitination in cells and in vitro

Because we find a dependence on TRAF6 for INAVA puncta formation and IL-1β inflammatory responses, we hypothesized that INAVA may function in the TRAF6 ubiquitin ligase cascade essential for IL-1β signaling (*Xia et al., 2009*). To test this, we examined protein polyubiquitination in HEK293T cells expressing HA-tagged ubiquitin. Cells expressing both TRAF6 and INAVA exhibited much higher levels of protein ubiquitination as assessed by anti-HA immunoblot (*Figure 5A*, compare lane 4 with lanes 1–3). Cells expressing the short INAVA-S isoform lacked this response

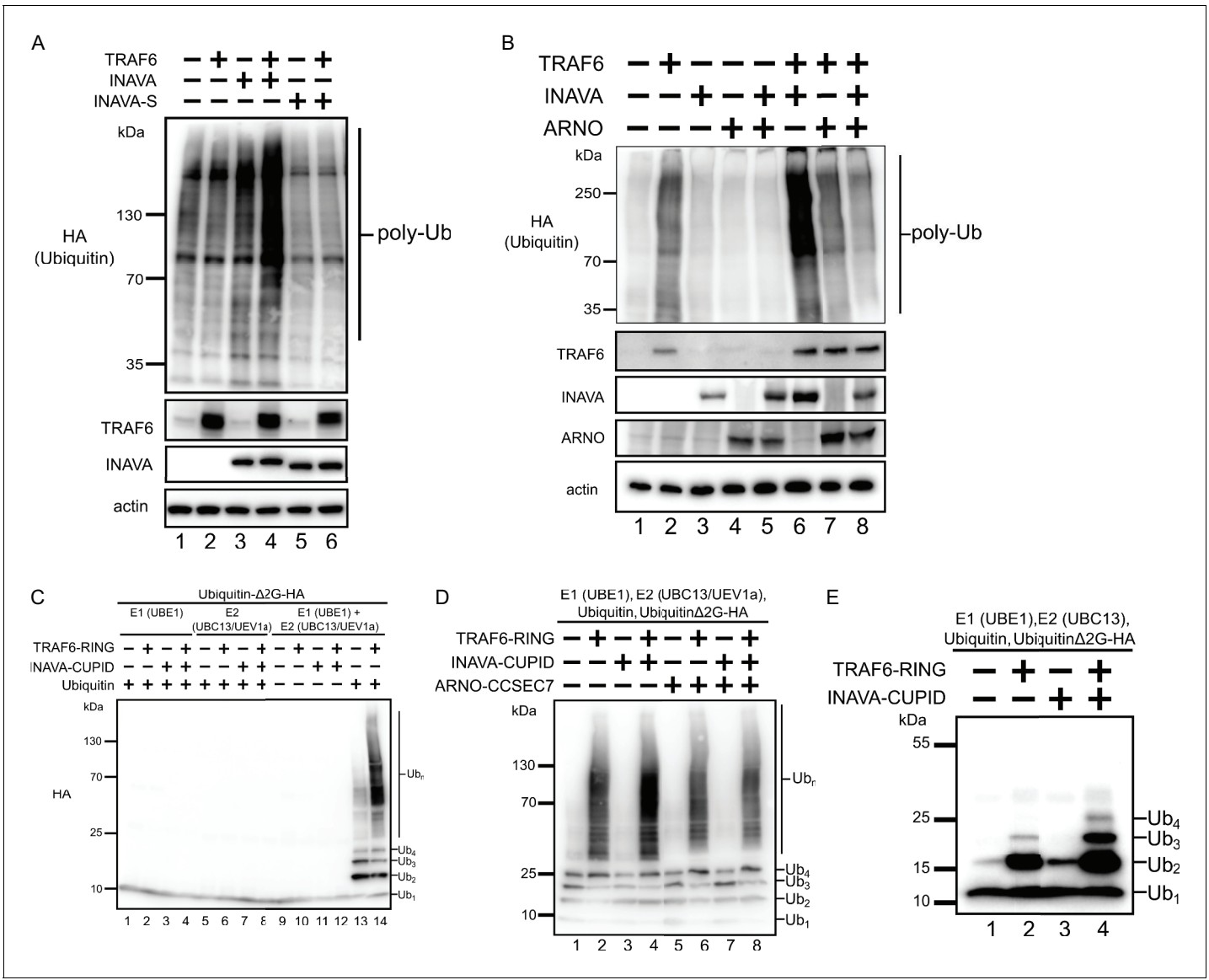

**Figure 5.** INAVA enhances TRAF6 dependent polyubiquitination in cells and in vitro. (**A**) HEK293T cells were transfected with HA-Ubiquitin and with or without myc-TRAF6. Whole cell lysates were prepared in RIPA buffer with protease inhibitors and analyzed by immunoblot for HA (ubiquitin), anti-TRAF6, and GFP (INAVA). Actin served as loading control. (**B**) HEK293T cells were transfected HA-Ubiquitin and indicated constructs. Whole cell lysate was prepared as above and analyzed by immunoblot for anti-myc (myc-INAVA 74 kDa, myc-ARNO 47 kDa), anti-TRAF6, and actin as loading control. (**C**) Cell-free reconstitution of polyubiquitination requires recombinant E1 (UBE1), E2 (UBC13/UEV1a), E3 (TRAF6-RING aa50-211), wild type Ubiquitin and pseudo-substrate UbiquitinΔ2G-HA. Reactions were incubated at 25°C for 2 hr and analyzed by immunoblot with anti-HA to detect ubiquitination on the pseudo-substrate. (**D**) In vitro ubiquitination reaction with recombinant MBP-INAVA-CUPID and ARNO CC-SEC7 along with E1 (UBE1), E2 (UBC13/UEV1a), E3 (TRAF6-RING domain aa 50–211), and pseudo-substrate UbiquitinΔ2G-HA. Reactions were incubated at 25°C for 2 hr and analyzed by immunoblot with HA to detect ubiquitination on the pseudo-substrate. (**E**) In vitro ubiquitination reaction as in (**D**) except with only Ubc13 as E2.
DOI: https://doi.org/10.7554/eLife.38539.014

(*Figure 5A*, compare lanes 5 and 6 with lane 4), consistent with the evidence that INAVA-S cannot form cytosolic puncta or mediate IL-1β inflammatory responses. We also found that overexpression of ARNO strongly inhibited INAVA-TRAF6 induced ubiquitination (*Figure 5B*, compare lane 8 with lane 6), consistent with our discovery that ARNO can act as a negative regulator in this pathway. Thus, INAVA appears to act as an enhancer of the ubiquitin-cascades induced by TRAF6, and this is blocked by ARNO.

To show INAVA is minimally sufficient for this effect, we reconstituted these reactions in vitro using recombinant purified proteins. For these studies, we produced recombinant HA-tagged ubiquitin lacking the terminal two glycines (UbiquitinΔ2G-HA), which we used as pseudosubstrate to ensure specificity for the ubiquitination-reactions. Because the di-glycine motif at the C-terminus of ubiquitin is required for covalent attachment to target lysines, the UbiquitinΔ2G-HA pseudosubstrate cannot be attached to other molecules added in the in vitro reactions, and the only detectable product can be ubiquitination of the pseudosubstrate. Control studies showed that the reactions reconstituted with wild type ubiquitin, UBE1 (E1), heterodimer UBC13/UEV1a (E2), and the E3 ligase TRAF6-RING (aa 50–211) produced polyubiquitinated UbiquitinΔ2G-HA (*Figure 5C*, compare lane 14 with all others). Reactions lacking either E1, heterodimer E2, E3, or ubiquitin yielded no product. When the same reactions were reconstituted with the addition of INAVA-CUPID, the polyubiquitination of UbiquitinΔ2G-HA was amplified (*Figure 5D*, compare lane 4 with lane 2). Reactions performed with just UBC13 as the E2 ligase were not as processive, but the amplification by INAVA-CUPID was still clearly apparent (*Figure 5E*, compare lanes 4 and 2). Thus, the INAVA-CUPID domain has activity as an enhancer of TRAF6 ubiquitination. This we propose explains the ability of full-length INAVA to promote inflammatory signaling by IL-1β and MDP-induced NOD2 activation in intact cells. Notably, the INAVA-dependent amplification of ubiquitination in vitro was fully inhibited by the addition of recombinant ARNO (*Figure 5D*, compare lane 8 with lane 4), explaining how ARNO can act by binding INAVA to negatively regulate the inflammatory response in this pathway.

## Discussion

INAVA-CUPID binds ARNO, targets the molecule to lateral membranes of epithelial monolayers, and enables ARNO to affect F-actin assembly without the need for GEF activity or positive ARF-GTP feedback (*Figure 2G*). This is a non-canonical mechanism for ARNO action, and one, we propose, explains how the INAVA-ARNO complex acts as a tunable factor for control of epithelial cell-cell contacts at mucosal surfaces. The impaired barrier integrity observed in INAVA knockout mice and cells can plausibly be explained by loss of INAVA-ARNO cortical actin promoting complexes at lateral membranes. This mechanism of action differs from the model proposed for INAVA mediated degradation of cytohesin 1 delineated in a recent report (*Mohanan et al., 2018*). It is possible that INAVA acts in different signaling complexes and operates in both ways depending on cell context. In our case, however, using both human epithelial cell lines and primary human macrophages, we find that cytohesins and INAVA cooperate to regulate downstream outcomes.

We also find that INAVA acts an enhancer in the ubiquitin cascades required for IL-1β signaling, and that CUPID is sufficient to reconstitute this activity in vitro. Thus, CUPID can functionally and directly act with the UBC13/UEV1 E2- and TRAF6 E3-ligases in polyubiquitination. The recent proteomic study by *Mohanan et al., 2018* of proteins co-isolated by INAVA immunoprecipitation implicated such an interaction between INAVA and another E3-ligase FBXW11 (SCF complex). But whether INAVA acted in protein ubiquitination of cytohesin 1 through FBXW11 was not conclusively tested (*Mohanan et al., 2018*).

In intact cells, the N-terminal region (aa 1–99) of INAVA is required for puncta formation and CUPID mediated enhancement of the ubiquitin ligases, as only the long INAVA isoform acts in the IL-1β inflammatory responses. INAVA-S is inactive. And it does not form cytosolic puncta, which we suggest represent protein complexes required for signal transduction (signalosomes). In macrophages, the C-terminal region of INAVA is also implicated in inflammatory signaling by mediating scaffolding for NOD2 interacting proteins (*Yan et al., 2017*).

We also note the signaling puncta formed in response to IL-1β require the relocalization of INAVA from lateral membranes to the cytosol. An analogous relocalization of INAVA, from cytosol to nucleus, was also observed in NOD2 activated macrophages (*Yan et al., 2017*), though the

functional consequences for INAVA in the nucleus have yet to be defined. Thus, CUPID action is likely site-specific, dictated by the N- and C-terminal regions of the protein flanking the CUPID domain.

Strikingly, in the case of inflammatory signaling, INAVA-CUPID is inhibited in the ubiquitination reactions by ARNO, thus, elucidating a previously unappreciated function for ARNO as a negative-regulator of inflammatory responses. The result also shows how ARNO can coordinate CUPID function as it bridges between barrier function and inflammation. Given INAVA's dual activities in IL-1β and MDP signaling cascades, and in F-actin assembly at lateral membranes, we can imagine INAVA (together with ARNO) innately reacting to environmental factors in the gut mucosa as a form of guard receptor (*Dangl and Jones, 2001*) for the management of intestinal homeostasis.

IL-1 signaling is negatively regulated by anti-inflammatory IL-1 receptor antagonist (IL-1Ra), a naturally occurring inhibitor of IL-1 signaling. IL-1Ra inhibits inflammation by binding and competing with IL-1β for the same surface IL-1 receptor (*Aksentijevich et al., 2009*; *Ashwood et al., 2004*; *Sanchez-Munoz et al., 2008*). Elevated levels and imbalance between IL-1 and IL-1 receptor antagonist (IL-1Ra) have been observed in patients with Crohn's disease and ulcerative colitis, and the imbalance of anti-inflammatory (IL-1Ra) and proinflammatory regulators (IL-1) are proposed to contribute to the chronic inflammation observed in IBD (*Li et al., 2004*; *Ludwiczek et al., 2004*; *McAlindon et al., 1998*; *Reinecker et al., 1993*).

In summary, our results delineate the molecular activities of CUPID and uncover unexpected and opposing roles of INAVA-ARNO complexes in the maintenance of epithelial barrier function at mucosal surfaces, and in inflammatory signaling for epithelial cells and macrophages. The two mechanisms defined are intrinsically related to the pathogenesis of IBD, and likely will apply as general rules for the CUPID domains of FRMD4A, FRMD4B, and CCDC120 - proteins implicated in neurite outgrowth, and in human cancer, Alzheimer's, celiac, and heart disease (*Cappola et al., 2010*; *Garner et al., 2014*; *Goldie et al., 2012*; *Lambert et al., 2013*; *Torii et al., 2014*; *Velcheti et al., 2017*).

# Materials and methods

## Key resources table

| Reagent type (species) or resource | Designation | Source or reference | Identifiers | Additional information |
|---|---|---|---|---|
| Antibody | Rat anti-Ecadherin | Sigma | U3254 | (1:1000) |
| Antibody | Rat anti-HA | Sigma | 11867423001 | (1:1000) |
| Antibody | Rabbit anti-HA | Cell Signaling | C29F4 | (1:1000) |
| Antibody | Rabbit anti-GFP | Sigma | G1544 | (1:1000) |
| Antibody | Mouse anti-b-actin | Sigma | A5441 | (1:5000) |
| Antibody | Mouse anti-ARNO | Sigma | SAB1404698 | (1:1000) |
| Antibody | anti-mouse HRP | Sigma | A4416 | (1:500) |
| Antibody | anti-rabbit HRP | Sigma | A6154 | (1:500) |
| Antibody | anti-rat HRP | Sigma | A9037 | (1:500) |
| Antibody | Rabbit anti-p65 | Cell Signaling | 3033 | (1:1000) |

*Continued on next page*

*Continued*

| Reagent type (species) or resource | Designation | Source or reference | Identifiers | Additional information |
|---|---|---|---|---|
| Antibody | Rabbit anti-phospho-p65 | Cell Signaling | 4764 | (1:1000) |
| Antibody | Mouse anti-myc | Cell Signaling | 9B11 | (1:1000) |
| Antibody | Mouse phospho-ERK | Cell Signaling | E10 | (1:1000) |
| Antibody | Mouse phospho-p38 | Cell Signaling | 3D7 | (1:1000) |
| Antibody | Mouse phospho-JNK | Cell Signaling | G9 | (1:1000) |
| Antibody | Rabbit phospho-IkBa | Cell Signaling | 14D4 | (1:1000) |
| Antibody | Mouse anti-Traf6 | Santa Cruz | sc-8409 | (1:500) |
| Antibody | DRAQ5 | ThermoFisher | 62251 | (1:1000) |
| Antibody | anti-mouse Alexa-647 | ThermoFisher | A21237 | (1:500) |
| Antibody | Mouse anti-ZO-1 | Thermo Fisher | 339100 | (1:1000) |
| Antibody | Mouse AntiT-TNF | BD Bioscience | MAb1; MAb11 | (1:1000) |
| Antibody | Rat anti-IL6 | BD Bioscience | MQ2-13A5; MQ2-39C3 | (1:1000) |
| Antibody | Mouse anti-IL8 | BD Bioscience | G265-5; G265-8 | (1:1000) |
| Antibody | Rat anti-IL10 | BD Bioscience | JES3-9D7; JES3-12G8 | (1:1000) |
| Antibody | Mouse anti-IL-1b | Thermo Fisher | CRM56; CRM57 | (1:1000) |
| Antibody | Mouse anti-IL12 | Thermo Fisher | C8.3; C8.6 | (1:1000) |
| Antibody | Rabbit anti-INAVA | Abcam | ab121945 | (1:1000) |
| Cell Line (*Homo sapiens*) | HEK293T | ATCC | | |
| Cell Line (*Homo sapiens*) | MCF7 | ATCC | | |
| Cell Line (*Homo sapiens*) | Caco2BBe (C2BBe1) | ATCC | | |
| Cell Line (*Homo sapiens*) | HCT8 | ATCC | | |
| Software, algorithm | ImageJ | ImageJ (https://imagej.nih.gov/ij/) | | |
| Software, algorithm | GraphPad Prism | Graphpad Prism (https://www.graphpad.com/scientific-software/prism/) | | |

*Continued on next page*

Continued

| Reagent type (species) or resource | Designation | Source or reference | Identifiers | Additional information |
|---|---|---|---|---|
| Software, algorithm | Slidebook | Intelligent Imaging Innovations (https://www.intelligent-imaging.com/) | | |
| Recombinant DNA reagent | INAVA | Harvard Plasmid Repository | | |
| Recombinant DNA reagent | Cytohesin1 | Harvard Plasmid Repository | | |
| Recombinant DNA reagent | ARNO | Harvard Plasmid Repository | | |
| Recombinant DNA reagent | GRP1 | Harvard Plasmid Repository | | |
| Recombinant DNA reagent | Cytohesin 4 | Harvard Plasmid Repository | | |
| Recombinant DNA reagent | TRAF6 | Harvard Plasmid Repository | | |
| Recombinant DNA reagent | pHAGE-CMV-MCS-IRES-ZsGreen | Harvard Plasmid Repository | | |
| Recombinant DNA reagent | pGEXTEV | Kim Orth, UT Southwestern | | |
| Recombinant DNA reagent | pLVX-Puro | Clontech | | |
| Recombinant DNA reagent | pLVX-EF1α-AcGFP-N1 | Clontech | | |
| Recombinant DNA reagent | pLVX-EF1α-AcGFP-C1 | Clontech | | |
| Recombinant DNA reagent | pET24a | Clontech | | |
| Recombinant DNA reagent | pET28a | Clontech | | |
| Recombinant DNA reagent | pAG413 Gal-YFP | Cammie Lesser, Mass Gen. Hospital | | |
| Recombinant DNA reagent | pAG413Gal-mcherry | Cammie Lesser, Mass Gen. Hospital | | |
| Recombinant DNA reagent | pag413Gal-μNS | Cammie Lesser, Mass Gen. Hospital | | |
| Recombinant DNA reagent | pTY-shRNA-EF1a-Puro2a-GFP | Yi Zhang, Harvard Medical School | | |
| Recombinant DNA reagent | SV40-Renilla | Promega | | |
| Recombinant DNA reagent | NF-κB luciferase | Jonathan Kagan, Boston Children's Hospital | | |
| Recombinant DNA reagent | pCW57.1 | Addgene, gift from David Root | | |
| Recombinant DNA reagent | psPAX2 | Addgene, gift from Didier Trono | | |
| Recombinant DNA reagent | pCMV-VSVG | Addgene, gift from Bob Weinberg | | |

*Continued*

| Reagent type (species) or resource | Designation | Source or reference | Identifiers | Additional information |
|---|---|---|---|---|
| Recombinant DNA reagent | UBE1 | Addgene, gift from Cynthia Wolberger | | |
| Recombinant DNA reagent | UBC13 | Addgene, gift from Cynthia Wolberger | | |
| Recombinant DNA reagent | UEV1a | Addgene, gift from Cheryl Arrowsmith | | |
| Recombinant DNA reagent | plentiC RISPRv2 | Addgene, gift from Feng Zhang | | |
| Recombinant DNA reagent | pcDNA3.0 | ThermoFisher | | |
| Biological sample (*Homo sapien*) | Human Peripheral Blood Mononuclear Cells | this study | | Donors recruited at Yale University |
| Peptide, recombinant protein | Ubiquitinꓝ 2G-HA-His6 | this study | | Recombinantly expressed in house |
| Peptide, recombinant protein | His-ΔN17ARF1 | this study | | Recombinantly expressed in house |
| Peptide, recombinant protein | GST-INAVA-CUPID | this study | | Recombinantly expressed in house |
| Peptide, recombinant protein | His-MBP-INAVA-CUPID | this study | | Recombinantly expressed in house |
| Peptide, recombinant protein | UBC13 | this study | | Recombinantly expressed in house |
| Peptide, recombinant protein | UEV1a | this study | | Recombinantly expressed in house |
| Peptide, recombinant protein | His-Cytohesin-CC-SEC7 | this study | | Recombinantly expressed in house |
| Peptide, recombinant protein | Traf6-RING-His6 | this study | | Recombinantly expressed in house |
| Peptide, recombinant protein | ubiquitin | Boston Biochem | | |
| Chemical compound, drug | MANT-GTP | Jena Bioscience | | |
| Chemical compound, drug | TRITC-phalloidin | American Peptide | | 92014A |
| Sequence-based reagent | gRNAs | this study | | Recombinantly expressed in house |
| Sequence-based reagent | shRNAs | this study | | Recombinantly expressed in house |
| Sequence-based reagent | qPCR-primers | this study | | Recombinantly expressed in house |
| Sequence-based reagent | genotyping primers | this study | | Recombinantly expressed in house |
| Strain, strain background (*Escherichia coli*) | Rosetta2D E3pLyS | EMD Millipore | | |
| Strain, strain background (*E.coli*) | Shuffle T7 Express | New England Biolabs | | |

*Continued on next page*

Continued

| Reagent type (species) or resource | Designation | Source or reference | Identifiers | Additional information |
|---|---|---|---|---|
| Other | HiTrap QHP | GE Healthcare | | |
| Other | Superdex 200 10/300 GL | GE Healthcare | | |

### INAVA genetic information

GeneID: 55765. INAVA constructs amino acid numbering follows NCBI reference sequence NP_060735.3 lNAVA isoform one and NP_001136041.1 for INAVA isoform 2.

### Patient recruitment and genotyping

Informed consent was obtained per protocol approved by the institutional review board at Yale University. Healthy individuals were recruited and rs7554511 genotyping was performed by Taqman (Life Technologies, Grand Island, NY).

### Plasmids

Template plasmids for INAVA, Cytohesin 1, ARNO, GRP1, Cytohesin 4, TRAF6 was obtained from Harvard Plasmid Repository. pGEX-TEV was a gift from Kim Orth (UT Southwestern). pHAGE-CMV-MCS-IRES-ZsGreen was obtained from Harvard Plasmid Repository. pLVX-Puro, pLVX-EF1α-AcGFP-N1, pLVX-EF1α-AcGFP-C1, pET24a, pET28a are form Clontech. Yeast protein interaction plasmids: pAG413Gal-YFP, pAG413Gal-mcherry and pag413Gal-μNS for PIP assays were a gift from Cammie Lesser (Mass. General Hospital). pTY-shRNA-EF1a-Puro2a-GFP was originally from Yi Zhang (Harvard Medical School). SV40-Renilla was from Promega and NF-κB luciferase was provided by Jonathan Kagan (Boston Children's Hospital). pCW57.1 (Addgene plasmid 41393) was a gift from David Root. psPAX2 (Addgene plasmid 12260) was a gift from Didier Trono. pCMV-VSVG (Addgene 8454) was a gift from Bob Weinberg. UBE1 (Addgene plasmid 34965) and UBC13 (Addgene plasmid 51131) was a gift from Cynthia Wolberger. UEV1a (Addgene plasmid 25619) was a gift from Cheryl Arrowsmith. plentiCRISPRv2 (Addgene plasmid 52961) was a gift from Feng Zhang.

### Antibodies

Antibodies anti-E-cadherin (U3254), anti-HA (11867423001) anti-GFP (G1544), anti-actin (A5441), anti-ARNO (), anti-mouse HRP (A4416), anti-rabbit HRP (A6154) and anti-rat HRP (A9037) were purchased from Sigma, unless noted. Antibodies anti-ZO-1 (339100), anti-mouse Alexa-647 (A21237) and DRAQ5 (62251) were purchased from ThermoFisher. Antibodies p65 (3033), phospho-p65 (4764) and anti-myc (2276S) were purchased from Cell Signaling, unless noted. Traf6 (sc-8409) antibody was purchased from Santa Cruz. TRITC-phalloidin (92014A) was obtain from AmericanPeptide.

### Cell culture

HEK293T and MCF7 were grown on DMEM 10% FBS with pen/strep. Caco2BBe were grown with DMEM 15% FBS with pen/strep. HEK293T, MCF7 and Caco2BBe were authenticated using Short Tandem Repeat (STR) analysis and tested negative for mycoplasma contamination. Human peripheral blood mononuclear cells (PBMCs) were prepared from peripheral blood using Ficoll-Paque (GE Dharmacon, Lafayette, CO), and monocytes then isolated and cultured for 1 week for monocyte derived macrophages (MDMs).

### Confocal and cell imaging

Brightfield images were taken using an inverted microscope coupled to a Nikon camera. Confocal images were taken using a spinning disk confocal head (CSU-X1, PerkinElmer, Boston, MA) coupled to an inverted Zeiss Axiovert 200M microscope (Carl Zeiss, Jena, Germany). Imaging system was operated using Slidebook (Intelligent Imaging Innovations, Denver, CO).

## Confocal imaging line scans

Caco2BBe WT cells were cocultured with stably expressed INAVA or INAVA-GFP and myc-ARNO constructs. Cells were grown on coverslips for 2 days, fix and stain as above. F-actin was stained with TRITC-phalloidin. ImageJ plot histogram function was used for line scans analysis. Line width was set at 10 pixels and the fluorescence F-actin intensity across cell-cell junctions. Peak positions were aligned based on the maximum peak value along the line.

## Immunocytochemistry

Cells plated either on glass coverslips or transwell filters were washed with PBS and fixed with 4% paraformaldehyde for 30 min. Cells were washed with PBS, permeabilize with PBS + 0.2% saponin and blocked with PBS + 0.2% saponin+5% normal goat for 1 hr. Primary antibodies were incubated for at least 1 hr at room temperature to overnight at 4°C. Cells were washed several times with PBS + 0.2% saponin and then incubated for 30 min with Alexa Fluor secondary antibodies, TRITC-phalloidin and DRAQ5 where indicated. Cells were washed and mounted on to slides.

## Protein purification

Plasmids were transformed into Rosetta2DE3pLys (EMD Millipore) and grown on 2XYT (BD) plates with appropriate antibiotics. Bacterial colonies were scrapped and inoculated onto starter liquid culture of 2XYT + antibiotics and grown at 37°C for 3–5 hr. The 5–10 ml starter culture were inoculated into large scale flasks containing 2XYT + antibiotics flasks and grown to OD600 = 0.8–1.0. Cultures were induced with 0.4 mM IPTG overnight at room temperature. Cells were harvested resuspended in 10 mM Tris pH8.0+150 mM NaCl 0.5 mM Triton-X100 +1 mM PMSF and lysed with an Emulsiflex-5 (Avestin). Cells were spun 16 kg for 10 min and supernatant transfer to Cobalt beads (Goldbio.com) or Glutathione resin (Goldbio.com) depending if the protein were His-tagged or GST-tagged, respectively. HiTrap QHP (GE Life Sciences) buffer A = 20 mM Tris pH 8.5 and buffer B = 20 mM Tris pH 8.5 + 1M NaCl. Superdex 200 10/300 GL buffer = 20 mM Tris pH 7.5, 100 mM NaCl. UBC13-UEV1a was purified as a complex where the plasmids were cotransformed and coexpressed. UBC13-UEV1a complex was purified by cobalt purification and then onto HiTrap QHP. Cytohesin constructs were purified by cobalt and onwards to HiTrap QHP where bound fractions were collected. GST-INAVA-CUPID (pGEX-TEV) and His-MBP-INAVA-CUPID (pET24a) constructs were purified by glutathione or cobalt respectively and onwards to HiTRAP QHP where bound fractions were collected. GST-INAVA-CUPID+His-Cytohesin-CC-SEC7 (pET28a) complex were coexpressed in Rosetta2-DE3pLlys and purified by tandem cobalt to glutathione resin and then onto Hi-Trap QHP. Complex fractions were collected. Traf6-His (aa 50–211) in pET24a was transformed into SHuffle T7 Express (NEB). Starter culture, inoculating of large culture and induction with IPTG was performed as above with addition of 50 µM zinc chloride added to the expression media. Purification of Traf6-His6 (aa 50–211) was purified with cobalt purification and samples were further purified by size exclusion Superdex 200. UbiquitinΔ2G-HA-His6 (pET24a) was expressed in Rosetta2DE3pLys, expressed and purified with cobalt and Superdex200. His-ΔN17ARF1 was purified by cobalt resin and chelated with 2 mM EDTA. GDP nucleotide was added at 20 molar excess of His-ΔN17ARF1 protein concentration along with 60 mM $MgCl_2$. Protein was then dialyzed to 10 mM Tris pH 8, 150 mM NaCl +5 mM $MgCl_2$. All proteins were flash frozen with liquid nitrogen and stored in aliquots at −80°C.

## Guanine exchange assays

Recombinant HisARNO-CC-SEC7 alone or complexed with GST-INAVA-CUPID (1 µM) were incubated with MANT-GTP (25 µM) and ΔN17ARF1-GDP (40 µM) for 15 min at 25°C. Guanine exchange reactions were measured by fluorescence at ex/em 355/445 nm using a TECAN fluorescent plate reader. Background from control samples with MANT-GTP and ΔN17ARF1-GDP was subtracted.

## Knockdown and CRISPR knockout generation

shRNAs were cloned into pTY-shRNA-EF1a-Puro2a-GFP. Guides were cloned into plentiCRISPRv2 and lentivirus was generated in HEK293T cells by cotransfection with psPAX2 and pVSVG by polyethyleneimine (PEI) transfection. For each INAVA knockout line, two unique pair of guides were designed to target the introns that flank exon 2 of INAVA which knockout both isoforms. Lentivirus that target the 5-prime and 3-prime ends were pooled and co-transduced into Caco2BBe cells. Cells

were selected with puromycin 5 µg/ml and single cell cloned into 96-well plates by manual pipetting. Caco2BBe INAVA knockout lines were screen by genomic PCR, clones were genotyped by TA cloning with pGEM-T (Promega) and individual bacterial colonies were sequenced. TRAF6 knockout line in HEK293T cells were generated with two unique pair of guides targeting exon two with lentiviral transduction. Knockout clones were screened by genomic PCR and functionally by NF-κB luciferase reporter. HCT8 TRAF6 knockout line were generated with same guides are above. HCT8 cells tend not to have exon removal upon introduction of 2 pair of guides and so knockout clones were screened using downstream NF-κB phospho-p65 immunoblot upon IL-1β treatment at 10 ng/ml for 15 min.

| | |
|---|---|
| INAVA shRNA-1 Fwd | tttgctatgcttctgctctgagaattcaagagatctcagagcagaagcatagctttta |
| INAVA shRNA-1 Rev | agcttaaaaagctatgcttctgctctgagaatctcttgaattctcagagcagaagcatag |
| INAVA shRNA-2 Fwd | tttgtgcaagtctttgtacctgaattcaagagattcaggtacaaagacttgcactttta |
| INAVA shRNA-2 Rev | agcttaaaaagtgcaagtctttgtacctgaatctcttgaattcaggtacaaagacttgca |
| Control shRNA-Fwd | tttgagtacttcgaaatgtccgttttcaagagaaacggacatttcgaagtactctttta |
| Control shRNA-Rev | agcttaaaaagagtacttcgaaatgtccgtttctcttgaaaacggacatttcgaagtact |
| INAVA qPCR Fwd | caccttgccagcggagtatc |
| INAVA qPCR Rev | cctgcctgctgaggttctc |
| | |
| INAVA KO guide-1 Fwd1 | caccgggggcaggagattctggcac |
| INAVA KO guide-1 Rev1 | aaacgtgccagaatctcctgcccccc |
| INAVA KO guide-1 Fwd2 | caccgggaccagggactagggacta |
| INAVA KO guide-1 Rev2 | aaactagtccctagtccctggtccc |
| INAVA KO guide-2 Fwd1 | caccgaagctagatatgaaccatct |
| INAVA KO guide-2 Rev1 | aaacagatggttcatatctagcttc |
| INAVA KO guide-2 Fwd2 | caccggagccctcaggcactagggc |
| INAVA KO guide-2 Rev2 | aaacgccctagtgcctgagggctcc |
| INAVA KO genotyping Fwd | tataggtgtgacaacacagaggc |
| INAVA KO genotyping Rev | gatggacacagcaaagtccctat |
| TRAF6 KO guide Fwd1 | caccgtttgtttctgttagggatgc |
| TRAF6 KO guide Rev1 | aaacgcatccctaacagaaacaaac |
| TRAF6 KO guide Fwd2 | caccgacgtgagattctttctctga |
| TRAF6 KO guide Rev2 | aaactcagagaaagaatctcacgtc |
| TRAF6 KO genotyping Fwd | tataggtgtgacaacacagaggc |
| TRAF6 KO genotyping Rev | gatggacacagcaaagtccctat |

## Calcium chelation

Caco2BBe WT and stable cells were grown on 0.4 µM polycarbonate filters for 10–14 days. 6 mM EGTA final concentration was added to basolateral chamber for 3 min. Cells were immediately washed with PBS and fixed with 4% PFA for 30 min. Cells were stained with E-cadherin and ZO-1 and Alexa Fluor secondary.

## Low calcium assay

Caco2BBe lines were grown on 24-well 0.4 µM polycarbonate transwell filters were grown at least 1 week. Cells were washed and grown in DMEM calcium free media, supplemented with 50 µM calcium +10% FBS. Transepithelial electrical resistance was measured before and 24 hr after low calcium treatment.

## Permeability assays

Caco2BBe lines were grown on 24-well 0.4 µM polycarbonate transwell filters for three days. Cells were washed and changed to DMEM + 10% FBS no phenol red media. 1 mg/ml FITC-dextran 4 Da was added apically. Cells were incubated for 2 hr at 37°C + 5% CO2. Basolateral media was collected and read in a 6-well plate reader ex/em 490/525 nm.

## qRT-PCR

RNA was isolated by QIAshredder and RNeasy Mini kit (Qiagen), reverse transcribed with SuperScript II (ThermoFisher) and qPCR by SYBR Green (BioRad) on a CFX96 (BioRad). Samples were normalized to actin.

## Luciferase assays

Firefly luciferase was assayed using Bright-Glo (Promega) or with homemade luciferase reagent as stated previously (*Baker and Boyce, 2014*). Renilla luciferase reagent was made and assayed as stated previously (*Baker and Boyce, 2014*).

## MDM stimulation

Cultured MDMs were treated with MDP (Bachem, Torrance, CA) or recombinant human IL-1β (BioLegend, San Diego, CA). Supernatants were assayed for TNF (clones MAb1 and Mab11), IL6 (clones MQ2-13A5 and MQ2-39C3), IL8 (clones G265-5 and G265-8), IL10 (clones JES3-9D7 and JES3-12G8) (BD Biosciences, San Jose, CA) or IL1β (clones CRM56 and CRM57) and IL12 (clones C8.3 and C8.6) (ThermoFisher, Waltham, MA) by ELISA.

## Transfection of DNA vectors in MDMs

INAVA was subcloned from BC106877 (ORD3016) plasmid (Transomic, Huntsville, AL) into pcDNA3.0 along with an HA tag. HA-INAVA CUPID deletion (CUPID Δ) and CUPID only variants were generated through site-directed mutagenesis (QuikChange Lightning Kit; Agilent Technologies). For plasmid transfection, 4.5 µg empty vector (pcDNA3.0) or INAVA variants, or the indicated concentrations of ARNO vector were transfected into MDMs by Amaxa nucleofector technology.

## Phosphoprotein and total protein detection in MDMs

Intracellular proteins were detected in permeabilized MDMs by flow cytometry with Alexa Fluor 488-, phycoerythrin-, Alexa Fluor 647- or biotin-labeled antibodies to phospho-ERK (clone E10), phospho-p38 (clone 3D7), phospho-JNK (clone G9), phospho-IκBα (clone 14D4), HA (clone C29F4), Myc (clone 9B11) (Cell Signaling Technology, Danvers, MA) or INAVA (Abcam, Cambridge, MA).

## Cells-based polyubiquitination assays

HEK293T wild type or stably expressed INAVA-GFP constructs were transfected with HA-Ubiquitin and where indicated myc-INAVA, myc-TRAF6 or FLAG-TRAF6 and myc-ARNO constructs by PEI transfection method between 16–20 hr. Stable INAVA-GFP lines had better comparable INAVA expression to each other than transient transfection. Cells were washed two times with PBS and lysed in RIPA +2 mM EDTA +protease inhibitors. Lysate were spun 20 kg for 5 min and supernatant was collected. Sample buffer was added and heated for 10 min at 95°C. Samples were load loaded onto SDS-PAGE gels and immunoblotted with anti-HA and detected by HRP using SuperSignal West Femto (Fisher Scientific). Images were taken with the Azure c300 system.

## Cell-free in vitro ubiquitination assay

Ubiquitination assay were performed as follows: UBE1 (250 nM), heterodimer UBC-UEV1a (1 µM), TRAF6-RING (2 µM), His-MBP-CUPID (350 nM), ARNO-CC-SEC7 (4 µM), wild type ubiquitin (15 µM,

Boston Biochem) and UbiquitinΔ2G-HA (5 μM) were incubated at 25°C for 2 hr in buffer 25 mM HEPES pH 7.5, 10 mM NaCl, 2 mM ATP, 7 mM MgCl2, 1 mM DTT. Reactions with UBC13 as the E2 enzyme used 15 μM protein along with increased TRAF6-RING (4 μM) and His-MBP-CUPID (2 μM). Ubiquitination reactions were stopped with sample buffer. Samples were load loaded onto SDS-PAGE gels and immunoblotted and imaged as above.

## Statistical analysis

Significance was assessed using two-tailed t-test or two-way ANOVA where indicated $p < 0.05$ was considered significant. Lines over multiple bars indicate same significance values for these bars. Graphs were generated using GraphPad Prism.

## Acknowledgements

We thank the members of the Lencer Lab for critical reading and discussions. Marian Neutra for critical reading of the manuscript. Scott B Snapper for discussion and reagents. This work is supported by the Crohn's and Colitis Foundation and Rubin-Wolpow Fellowship. National Institute of Health grants T32HD007466 (PL), DK099097 (CA), DK048106, DK084424, and the Harvard Digestive Diseases Center P30 DK034854 (Core C) (WIL).

# Additional information

## Competing interests

Xiaomo Jiang: is affiliated with Novartis. The author has no other competing interests to declare. The other authors declare that no competing interests exist.

## Funding

| Funder | Grant reference number | Author |
| --- | --- | --- |
| National Institutes of Health | T32HD007466 | Phi H Luong |
| Crohn's and Colitis Foundation of America | Career Development Award | Phi H Luong |
| Boston Children's Hospital | Rubin-Wolpow Fellowship | Phi H Luong |
| National Institutes of Health | DK099097 | Clara Abraham |
| National Institutes of Health | DK048106 | Wayne I Lencer |
| National Institutes of Health | DK084424 | Wayne I Lencer |
| National Institutes of Health | P30 DK034854 | Wayne I Lencer |

The funders had no role in study design, data collection and interpretation, or the decision to submit the work for publication.

## Author contributions

Phi Luong, Conceptualization, Resources, Validation, Investigation, Methodology, Writing—original draft, Project administration, Writing—review and editing; Matija Hedl, Jie Yan, Conceptualization, Resources, Formal analysis, Validation, Investigation, Methodology, Writing—review and editing; Tao Zuo, Conceptualization, Validation, Investigation, Methodology, Writing—review and editing; Tian-Min Fu, Jay R Thiagarajah, Steen H Hansen, Cammie F Lesser, Hao Wu, Conceptualization, Resources, Methodology, Writing—review and editing; Xiaomo Jiang, Conceptualization, Methodology, Writing—original draft, Writing—review and editing; Clara Abraham, Conceptualization, Resources, Formal analysis, Funding acquisition, Validation, Investigation, Methodology, Writing—original draft, Writing—review and editing; Wayne I Lencer, Conceptualization, Resources, Supervision, Funding acquisition, Methodology, Writing—original draft, Project administration, Writing—review and editing

## Author ORCIDs

Phi Luong http://orcid.org/0000-0003-1642-7415
Wayne I Lencer http://orcid.org/0000-0001-7346-2730

## Decision letter and Author response

Decision letter https://doi.org/10.7554/eLife.38539.017
Author response https://doi.org/10.7554/eLife.38539.018

## Additional files

### Supplementary files

• Transparent reporting form
DOI: https://doi.org/10.7554/eLife.38539.015

### Data availability

All data analysed during this study are included in the manuscript. Source data have been provided for Figure 3 C-F.

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
