## [Decision Letter]

Thank you for submitting your article "INAVA-ARNO complexes bridge mucosal barrier function with inflammatory signaling" for consideration by *eLife*. Your article has been reviewed by two peer reviewers and the evaluation has been overseen by Tadatsugu Taniguchi as the Senior Editor. The reviewers have opted to remain anonymous.

The reviewers have discussed the reviews with one another and the Reviewing Editor has drafted this decision to help you prepare a revised submission.

Summary:

In this study, the authors analyzed the function of INAVA. First, the authors generated INAVA KO Caco2BBe cells, and showed the role of INAVA in the maintenance of epithelial barrier integrity. Then, the authors analyzed the mechanisms by which INAVA regulates barrier function. As previously reported, INAVA interacted with cytohesins, such as ARNO. CUPID domain of INAVA associated with coiled-coil domain of ARNO. They further showed that INAVA stabilizes ARNO expression by associating with it. The association of INAVA and ARNO mediated recruitment of ARNO to the lateral membrane of epithelia in a GEF-independent manner.

Because ARNO has been shown to be involved in the IL-1β signaling, the authors next analyzed the effect of INAVA/ARNO in the IL-1β signaling, and found that ARNO negatively regulates the INAVA-dependent IL-1β signaling.

In the next experiment, the authors analyzed primary human macrophages isolated from individuals carrying the INAVA-IBD risk allele. They found that INAVA promotes the IL-1β-induced inflammatory responses in macrophages. As the mechanism for the INAVA-dependent enhancement of the IL-1β signaling, they showed that INAVA enhanced ubiquitination of TRAF6, which was inhibited by ARNO.

Overall, this manuscript is well written, interesting, and will help resolve the debate in the field. We have the following suggestions to improve the manuscript.

Essential revisions:

1) What is most concerning is that the authors performed experiments with overexpression system alone. The authors should perform the following experiments to strengthen their interesting findings.

a) The authors should show expression of endogenous ARNO in wild-type and INAVA KO Caco2BBe cells (related to Figure 2).

b) The authors should analyze IL-1β responses in INAVA KO Caco2BBe cells (related to Figure 3).

c) The authors should compare the IL-1β responses of wild-type MDM and IBD-allele MDM (related to Figure 4).

d) The authors are better to show ARNO expression in epithelial cells of IBD patients with the INAVA gene SNP.

2) The authors should analyze lamina propria mononuclear cells in terms of the IL-1 responses, when samples are available.

3) The authors should at least discuss how the IL-1 response is implicated in the pathogenesis of IBD.

*Reviewer #1:*

INAVA (previously called C1ORF106) was identified as a risk factor of IBD. A previous study showed that INAVA regulates intestinal barrier integrity by stabilizing adherens junction.

In this study, the authors analyzed the function of INAVA. First, the authors generated INAVA KO Caco2BBe cells, and showed the role of INAVA in the maintenance of epithelial barrier integrity. Then, the authors analyzed the mechanisms by which INAVA regulates barrier function. As previously reported, INAVA interacted with cytohesins, such as ARNO. CUPID domain of INAVA associated with coiled-coil domain of ARNO. They further showed that INAVA stabilizes ARNO expression through the association. The association of INAVA and ARNO mediated recruitment of ARNO to the lateral membrane of epithelia in a GEF-independent manner.

Because ARNO has been shown to be involved in the IL-1β signaling, the authors next analyzed the effect of INAVA/ARNO in the IL-1β signaling, and found that ARNO negatively regulates the INAVA-dependent IL-1β signaling.

In the next experiment, the authors analyzed primary human macrophages isolated from individuals carrying the INAVA-IBD risk allele. They found that INAVA promotes the IL-1β-induced inflammatory responses in macrophages. As the mechanism for the INAVA-dependent enhancement of the IL-1β signaling, they showed that INAVA enhanced ubiquitination of TRAF6, which was inhibited by ARNO.

What is most concerning in this study is that the authors show the contradictory results to the pervious study by using overexpression system alone. The previous study showed INAVA induced degradation of cytohesin using gene-targeted KO mice. However, this study just used INAVA over-expressing cells to show that INAVA stabilizes cytohesin expression.

1) In Figure 2B, the authors showed that INAVA stabilized ARNO expression. However, these are shown in an overexpression system alone. The authors should show expression of endogenous ARNO in wild-type and INAVA KO Caco2BBe cells.

2) The authors showed that INAVA mediates ARNO localization to lateral membranes. However, they did not show the mechanisms by which ARNO regulates cell-cell contact and barrier functions.

3) In Figure 3, again, the authors used only INAVA-overexpressing cells for the analysis of IL-1β responses. The authors should analyze IL-1β responses in INAVA KO Caco2BBe cells.

4) In Figure 4, the reviewer wonders why the authors did not compare the IL-1β responses of wild-type MDM and IBD-allele MDM. Again, the authors used INAVA-overexpressing cells alone.

*Reviewer #2:*

In this paper, Luong et al. describe a bifunctional role for the IBD-risk gene, IVANA, in the crosstalk between the environment and barrier function. Both activities require CUPID, which binds to the cytohesin, ARF-GEF ARNO, to affect lateral membrane F-actin assembly underlying cell-cell junctions and barrier function. Unexpectedly, when bound to CUPID, ARNO affects F-actin dynamics in the absence of its canonical activity as a guanine nucleotide-exchange factor. Upon exposure to IL-1β, INAVA relocates to form cytosolic puncta, where CUPID amplifies TRAF6-dependent polyubiquitination and inflammatory signaling. In this case, ARNO binding to CUPID negatively-regulates polyubiquitination and the inflammatory response. INAVA and ARNO act similarly in primary human macrophages responding to IL-1β and NOD2 agonists. The authors conclude that, INAVA-CUPID exhibits dual functions, coordinated directly by ARNO, that bridge epithelial barrier function with extracellular signals and inflammation.

This is a well executed piece of work investigating the mechanism of action of a relevant IBD risk gene, namely INAVA. The experiments are technically well done, and the five figures show data of high quality, reporting mechanistic results. I have, however, three concerns:

1) The authors use Caco2BEE cells as a model of epithelial cells. It is well known that the cells are cancer cells with questionable relevance to normal epithelials cells, especially in relationship to inflammatory bowel disease (IBD).

2) The authors use macrophages derived from peripheral blood cells of patients carrying the homozygous mutation of INAVA. It is well known that peripheral blood cells are very different from tissue macrophages. There is no attempt to use in this study, for example, lamina propria mononuclear cells, from these patients.

3) The authors investigate the effects of IL-1β as a prototypic proinflammatory cytokine to establish a link between inflammation and barrier function. The role of IL-1β in IBD remains controversial, and some evidence actually suggests that IL-1β has an anti-inflammatory effect on the epithelium. It is somehow disappointing that no effect(s) were observed with TNFα, which is an established proinflammatory cytokine in IBD.

---

## [Author Response]

Essential revisions:1) What is most concerning is that the authors performed experiments with overexpression system alone. The authors should perform the following experiments to strengthen their interesting findings.

These comments were most helpful. All studies were completed, except for studies requiring new human biopsy samples, and results are consistent with our original interpretations.

a) The authors should show expression of endogenous ARNO in wild-type and INAVA KO Caco2BBe cells (related to Figure 2).

These new experiments were performed and appear in Figure 2—figure supplement 1E and F of the revised manuscript. The results show that over-expression of INAVA-GFP increases the protein levels of ARNO and in 2 independently created INAVA KO Caco2BBe cell lines we find no evidence for increased ARNO expression levels.

The text has been revised in subsection “CUPID-domain directly interacts with cytohesin family members”:

“Two independently created INAVA KO Caco2BBe cell lines showed no evidence for increased ARNO levels (Figure 2—figure supplement 1F), unlike the results recently reported for INAVA’s effect on cytohesin 1, a closely related ARF-GEF member of the cytohesin family.”

b) The authors should analyze IL-1β responses in INAVA KO Caco2BBe cells (related to Figure 3).

These new experiments were performed and appear in Figure 3—figure supplement 1C of the revised manuscript. They show that INAVA is not required for IL-1β signaling in the Caco2BBe cells. Rather INAVA acts to amplify the response – as delineated in the paper.

The text has been revised in subsection “INAVA-ARNO regulates IL-1β signalling”:

“We conclude INAVA-puncta mark the formation of multiprotein signalosomes required for affecting inflammatory responses. […] The signalosomes mediating this activity depend on the INAVA-N-terminal and CUPID domains (Figure 3D), and on the enzymatic activity of TRAF6.”

c) The authors should compare the IL-1β responses of wild-type MDM and IBD-allele MDM (related to Figure 4).

These new studies on human MDMs were performed and appear in Figure 4 of the revised manuscript. They show that the INAVA (low expression) risk allele (CC) has attenuated signal transduction compared to the INAVA (high expression) allele (AA).

The text has been revised in subsection “INAVA-ARNO amplifies inflammatory signaling in human macrophages”:

“To further address the relevance of these findings, we prepared primary monocyte-derived human macrophages (MDMs) from individuals carrying the INAVA-IBD rs7554511 CC risk allele. […] And consistent with our in vitro results, both IL-1β-induced MAPK and NF-κB signaling (Figure 4B), and cytokine secretion (Figure 4C) were reduced in MDMs from rs7554511 CC IBD risk carriers.”

d) The authors are better to show ARNO expression in epithelial cells of IBD patients with the INAVA gene SNP.

These studies were not performed as they would require us to perform additional endoscopy and colonoscopy of patients harboring the high expression and low expression (risk) alleles. This is experimentally/financially prohibitive and we lack IRB approval. Furthermore, justification would be based only upon research needs – there would be no benefit to the patient and this requires an invasive study.

2) The authors should analyze lamina propria mononuclear cells in terms of the IL-1 responses, when samples are available.

We did not perform these studies. As explained above, this would require an additional invasive procedure without medical justification. Going forward, we will make attempts to do these studies whenever appropriate, and we will prepare intestinal organoids for the biochemical studies suggested above.

3) The authors should at least discuss how the IL-1 response is implicated in the pathogenesis of IBD.

Reviewer 2 further writes: “The authors investigate the effects of IL-1β as a prototypic proinflammatory cytokine to establish a link between inflammation and barrier function. The role of IL-1β in IBD remains controversial, and some evidence actually suggests that IL-1β has an anti-inflammatory effect on the epithelium. It is somehow disappointing that no effect(s) were observed with TNFα, which is an established proinflammatory cytokine in IBD.”

This was a helpful suggestion. IL-1β is one of the most potent inflammatory cytokines. The text has been revised to briefly discuss the evidence for IL-1β in the pathogenesis of IBD. But we do emphasize that our work does not address, or make the claim, that IL-1β underlies the basis of disease.

With respect to TNFα, we do find this cytokine induces the same phenotype, but in our HCT8 intestinal cell model (where the punctae phenotype is stronger compared to Caco2BBe) the effect of inducing INAVA punctae by TNFα is smaller (Figure 3—figure supplement 1B).

The text has been revised in the Discussion section:

“IL-1 signaling is negatively regulated by anti-inflammatory IL-1 receptor antagonist (IL-1Ra), a naturally occurring inhibitor of IL-1 signaling. IL-1Ra inhibits inflammation by binding and competing with IL-1β for the same surface IL-1 receptor (Aksentijevich et al., 2009; Ashwood et al., 2004; Sanchez-Muñoz et al., 2008). Elevated levels and imbalance between IL-1 and IL-1 receptor antagonist (IL-1Ra) have been observed in patients with Crohn’s disease and ulcerative colitis, and the imbalance of anti-inflammatory (IL-1Ra) and proinflammatory regulators (IL-1) are proposed to contribute to the chronic inflammation observed in IBD (Li et al., 2004; Ludwiczek et al., 2004; McAlindon et al., 1998; Reinecker et al., 1993).”